# Magnetic Resonance Image Quality Assessment by Using Non-Maximum Suppression and Entropy Analysis

**DOI:** 10.3390/e22020220

**Published:** 2020-02-16

**Authors:** Rafał Obuchowicz, Mariusz Oszust, Marzena Bielecka, Andrzej Bielecki, Adam Piórkowski

**Affiliations:** 1Department of Diagnostic Imaging, Jagiellonian University Medical College, 19 Kopernika Street, 31-501 Cracow, Poland; rafalobuchowicz@su.krakow.pl; 2Department of Computer and Control Engineering, Rzeszow University of Technology, W. Pola 2, 35-959 Rzeszow, Poland; marosz@kia.prz.edu.pl; 3Faculty of Geology, Geophysics and Environmental Protection, AGH University of Science and Technology, al. Mickiewicza 30, 30-059 Cracow, Poland; 4Faculty of Electrical Engineering, Automation, Computer Science and Biomedical Engineering, AGH University of Science and Technology, al. Mickiewicza 30, 30-059 Cracow, Poland; azbielecki@gmail.com; 5Department of Biocybernetics and Biomedical Engineering, AGH University of Science and Technology, al. Mickiewicza 30, 30-059 Cracow, Poland; pioro@agh.edu.pl

**Keywords:** blind image quality assessment, magnetic resonance images, entropy, non-maximum suppression

## Abstract

An investigation of diseases using magnetic resonance (MR) imaging requires automatic image quality assessment methods able to exclude low-quality scans. Such methods can be also employed for an optimization of parameters of imaging systems or evaluation of image processing algorithms. Therefore, in this paper, a novel blind image quality assessment (BIQA) method for the evaluation of MR images is introduced. It is observed that the result of filtering using non-maximum suppression (NMS) strongly depends on the perceptual quality of an input image. Hence, in the method, the image is first processed by the NMS with various levels of acceptable local intensity difference. Then, the quality is efficiently expressed by the entropy of a sequence of extrema numbers obtained with the thresholded NMS. The proposed BIQA approach is compared with ten state-of-the-art techniques on a dataset containing MR images and subjective scores provided by 31 experienced radiologists. The Pearson, Spearman, Kendall correlation coefficients and root mean square error for the method assessing images in the dataset were 0.6741, 0.3540, 0.2428, and 0.5375, respectively. The extensive experimental evaluation of the BIQA methods reveals that the introduced measure outperforms related techniques by a large margin as it correlates better with human scores.

## 1. Introduction

The ubiquity of advancements in imaging has brought significant attention of medical specialists due to the role of the quality of displayed content in diagnosis [1,2,3]. The quality of Magnetic Resonance (MR) images depends on used hardware parts, software techniques, as well as human errors involving patient noncompliance or operator mistakes [4,5,6,7,8]. Therefore, the development of automatic image quality assessment (IQA) methods for MR scans is particularly important since the contamination of acquired images may compromise subsequent diagnosis and treatment. Moreover, such methods may support a selection of algorithms for image processing or parameters of imaging systems. Hopefully, a time-consuming examination of images by trained medical specialists can be avoided. Furthermore, the lack of reproducibility of subjective tests and personal quality preferences impeding scores of small groups encourages the use of automatic and repeatable IQA methods. The IQA measures are divided into three categories: Full-reference (FR), reduced-reference (RR), and no-reference or blind (BIQA) methods [9]. The full-reference methods compare input images with their non-distorted versions. However, most medical imaging systems do not produce pristine images, limiting the application range of FR methods [10]. The reduced reference techniques, in turn, require only a part of the information on the pristine image, and blind IQA methods assess images without any external information. Therefore, the development of BIQA approaches is desired.

Among the applications of FR-IQA methods to MR images, Baselice et al. [10] compared results of denoising approaches using Mean Square Error (MSE) with the Structural Similarity Index (SSIM) [11]. Jang et al. [12] employed the SSIM and the Root-Mean-Square Error (RMSE) for an evaluation of BIQA methods on synthetically distorted MR scans. In the work of Chow and Rajagopal [13], Noise Quality Measurement (NQM) with Feature SIMilarity (FSIM) were applied to evaluate a BIQA method. Recognition of a supportive role of FR measures in the assessment of medical images and the need for creating new datasets are among findings of that work [13]. The reduced-reference techniques are not used for the assessment of MR images. In the literature, several BIQA methods have been introduced. Interestingly, as the Signal-to-Noise Ratio (SNR) and Contrast-to-Noise Ratio (CNR) are frequently used for the assessment of medical images [14], they are often criticized due to the need of indication of clearly defined regions with tissue and background [14,15,16]. Considering MR images, Chow and Rajagopal [13] adapted Blind/Referenceless Image Spatial Quality Evaluator (BRISQUE) [17] by training it on MR images instead of natural images. The method fits the Mean Substracted Contrast Normalization (MSCN) of an image to the Generalized Gaussian Distribution (GGD). Similarly, the GGD was used by Jang et al. [12]. In that work, the characteristics of MR scans were taken into account by employing a multidirectional-filtering of images. In the work of Yu et al. [16], four BIQA methods, i.e., BRISQUE, Natural Image Evaluator (NIQE), Blind Image Integrity Notator using DCT statistics (BLIINDS-II), and Blind Image Quality Index (BIQI), were trained on the SNR scores. Their correlation with the SNR was investigated by Zhang et al. [18]. The BIQA methods for the assessment of brain MR scans were employed by Sandilya and Nirmala [19] and Osadebey et al. [20]. In the first work, the reconstructed scans were assessed with BRISQUE, while in the second approach, binary images of brain scans were evaluated considering noise, lightness, contrast, sharpness, and texture details.

The literature review reveals that the lack of BIQA approaches designed for MR scans is caused mainly by the lack of IQA databases of such images with subjective scores. Moreover, natural images differ from MR images concerning characteristics of used imaging systems for their registration, the complexity of captured structures, or noise. Considering the popularity of IQA methods designed for natural images, some of BIQA approaches for MR scans adapted or modified them. However, there exist many concepts in the IQA of natural images that are not yet utilized for MR images and they should be examined. Therefore, in this paper, apart from the novel BIQA approach designed for MR scans, a set of representative IQA methods is evaluated. Furthermore, a dataset with MR scans and subjective scores used in the evaluation is released.

The introduced method, ENtropy-based Magnetic resonance Image Quality Assessment measure (ENMIQA), takes into account thresholded local intensity differences obtained by using the non-maximum suppression (NMS) [21,22] operation and calculates the entropy of a sequence of extrema numbers. The extrema represent a set of filtered versions of an input image. Then, entropy is used for quality prediction.

The major contributions of this work are a novel method for the quality assessment of MR images and a comprehensive evaluation of the measure against the state-of-the-art IQA techniques on a dataset of MR images assessed by a large group of experienced radiologists.

The remainder of this paper is organized as follows. In Section 2, the approach is introduced. Then, in Section 3, it is evaluated against the related BIQA methods. Finally, in Section 4, the paper is concluded.

## 2. Proposed Image Quality Measure

In the introduced method, ENMIQA, an input image *I* is filtered to determine pixels that represent local intensity extrema. To determine which pixels should be selected, the NMS operation [21,22] is performed. However, to provide a more thorough examination instead of selecting pixels that are of greater or lesser intensity value than its surrounding neighbors, in this work, a sequence of intensity thresholds T=[1,2,⋯,S], $S\in Z^{+}$, is introduced. The NMS uses the threshold t∈T to indicate the local extrema. Consequently, image *I* for each threshold *t* is represented by the number of found local extrema I(t). This can be written as:(1)$$I\left(t\right)=\sum_{a=1}^{M}\sum_{b=1}^{N}T\left(a,b,t\right),$$
where a pair (a,b) denotes the pixel location within an image of the size M×N and T(a,b,t) is a test in which the NMS is calculated using the proposed threshold *t*. The test is obtained as follows:(2)$$T\left(a,b,t\right)=\left\{\begin{array}{cc} 1, & \mathrm{if}\;\forall_{\left(i,j\right)}I\left(a,b\right)>I\left(a+i,b+j\right)+t,\\ 1, & \mathrm{else}\;\mathrm{if}\;\forall_{\left(i,j\right)}I\left(a,b\right)<I\left(a+i,b+j\right)-t,\\ 0, & \mathrm{otherwise}, \end{array}\right.$$
where (i,j)∈{(0,1),(0,−1),(1,0),(−1,0)}. The pair of indices (i,j) forms the neighborhood of 3×3 pixels around the location (a,b). Finally, a sequence of sums I(T)=[I(t=1),I(t=2),⋯,I(t=S)] is obtained. Then, it is divided by the image size to normalize the values. To determine the quality of the input image *I*, entropy of I(T) is calculated.

Entropy is the fundamental concept of Shannon information theory [23,24]. It is usually considered in the framework of measure theory. Assuming that space *X* with a probabilistic measure μ and a countable partition P of *X* are given [25], the entropy *h* is:(3)$$h\left(\mu ,P\right)=\sum_{P\in P}s\left(\mu \left(P\right)\right),$$
where *s*: [0,1]→[0,∞) can be expressed as s(x)=−xlogx for 0<x≤1 and s(0)=0. Note that entropy equals zero if and only if there exists such P∈P that μ(P)=1. If *X* contains *R* elements, then $P=\{P_{1},\ldots ,P_{R}\}.$ Furthermore, if μ is based on counting measure, then Equation (Equation 3) has the following form:(4)$$h\left(\mu ,P\right)=-\sum_{i=1}^{R}k_{i}\log k_{i},$$
where $k_{i}=\frac{m_{i}}{m},$$m_{i}$ and *m* are the numbers of elements in $P_{i}$ and *X*, respectively. Entropy defined by Equation (Equation 4) reaches its maximum for the uniform distribution of the measure μ on the family P. Such defined entropy refers to the amount of information on (X,μ) introduced by P. Consequently, the inversely proportional relationship between entropy and information is often applied in practice.

In this paper, entropy analysis is used for the IQA of two-dimensional MR images. In such a context, it can be employed for measuring disorders. In MR scans of internal organs, single isolated impulses with higher or lower intensity concerning a local neighborhood are common in distorted images. Thus, the greater the value of the threshold *t* in the NMS, the greater the probability that the detected intensity irregularities are disorders that decrease the quality of an image. The observed discriminative capabilities of entropy regarding images of different qualities justify its use for the IQA of MR images. In this work, Equation (Equation 4) is directly used as a quality measure, assuming that a set *X* is expressed as {(I,t),t∈T} and T determines the partition of *X*. The main computational steps of the method are shown in Figure 1.

Figure 2 presents two MR images of different quality and the influence of *t* on the local extrema. As shown, the proposed method determines more extrema in images with more distortions.

## 3. Results and Discussion

In this section, a dataset that contains MR images with associated subjective scores is introduced. Then, the performance of ENMIQA against ten state-of-the-art related methods is evaluated using a typical methodology and discussed. Finally, the influence of parameters of ENMIQA on its performance is provided.

### 3.1. Experimental Data

The introduced ENMIQA and related techniques are evaluated on a dataset that contains MR images and subjective scores collected in tests with human subjects. The dataset consists of 70 T2-weighted MR images (T2w) extracted from the lumbar and cervical spine, brain, hip, knee, and wrist sequences in axial, sagittal, and coronal planes. The sequences were obtained for a group of 51 patients of 27–41 years old (26 men and 25 women). The study protocol was designed according to the guidelines of the Declaration of Helsinki and the Good Clinical Practice Declaration Statement. The data safety was ensured by removing the personal details from images. Written acceptance for conducting the study was obtained from the Ethics Committee of Jagiellonian University (no. 1072.6120.15.2017). To produce images with different quality for the IQA purposes, shortened sequences were acquired using Process Analytical Technology (PAT) I software (Siemens) and employing the GeneRalized Autocalibrating Partially Parallel Acquisitions (GRAPPA) 3 in which 25% of the echoes were acquired with 60% signal reduction regarding the original acquisition mode [26,27]. Then, images with distortion types that were not present in all examined body parts were rejected. The obtained dataset is characterized in Table 1. There are 15, 9, and 11 image pairs captured in sagittal, axial, and coronal planes, respectively. The size of the images ranges from 192×320 to 512×512. The subjective scores for images were obtained in a group of 31 experienced radiologists with more than six years of diagnostic reading residency. Each radiologist assessed two images of the same part of the body at once, spending a minute on the assessment of the pair. The images were scored from 1 to 5, with a higher score associated with better quality. The examination was repeated until all images in the dataset were assessed. Then, scores for images were averaged and the mean opinion score (MOS) was obtained. The number of radiologists that took part in the subjective tests was large enough to ensure that personal quality preferences do not impair the MOS. However, the number of images in the database depended on the number of medical professionals and the time spent on the examination. Exemplary images from the dataset can be seen in Figure 3.

### 3.2. Evaluation Methodology

According to the popular protocol for the performance evaluation of IQA measures, objective scores Q for images in a database are compared with subjective scores (i.e., MOS) S collected for them in tests with human subjects. Typically, the four criteria are used to characterize IQA measure [28]: Pearson correlation coefficient (PLCC), Spearman Rank order Correlation Coefficient (SRCC), Kendall Rank order Correlation Coefficient (KRCC), and Root Mean Square Error (RMSE). The PLCC and RMSE are calculated for the vector $Q_{p}$ obtained via a nonlinear mapping between objective scores Q and subjective scores S using fitted parameters of the regression model $\beta =[\beta_{1},\beta_{2},\cdots ,\beta_{5}]$, i.e., $Q_{p}=\beta_{1}\left(\frac{1}{2}-\frac{1}{1+\exp(\beta_{2}\left(Q-\beta_{3}\right))}\right)+\beta_{4}Q+\beta_{5}$. The PLCC is obtained as:(5)$$\mathrm{PLCC}\left(Q_{p},S\right)=\frac{{\bar{Q_{p}}}^{T}\bar{S}}{\sqrt{{\bar{Q_{p}}}^{T}\bar{Q_{p}}{\bar{S}}^{T}\bar{S}}},$$
where $\bar{Q_{p}}$ and $\bar{S}$ are mean-removed vectors. The SRCC is calculated as:(6)$$\mathrm{SRCC}\left(Q,S\right)=1-\frac{6\sum_{i=1}^{m}d_{i}^{2}}{m(m^{2}-1)},$$
where $d_{i}$ is the difference between *i*-th image in vectors of scores and *m* denotes the number of images in the dataset. The KRCC is obtained as:(7)$$\mathrm{KRCC}\left(Q,S\right)=\frac{m_{c}-m_{d}}{0.5m(m-1)},$$
where $m_{c}$, $m_{d}$ are the number of concordant and discordant pairs, respectively. The RMSE, in turn, is obtained as:(8)$$\mathrm{RMSE}\left(Q_{p},S\right)=\sqrt{\frac{{\left(Q_{p}-S\right)}^{T}\left(Q_{p}-S\right)}{m}}.$$

### 3.3. Comparative Evaluation

The ENMIQA is compared against the following ten related BIQA measures: SNRTOI [18], BPRI [29], ILNIQE [30], QENI [31], SISBLIM [32], metricQ [33], SSEQ [34], SINDEX [35], MEON [36], and DEEPIQ [37]. The SNRTOI [18] was implemented by authors of this paper, while other methods were run using their publicly available Matlab implementations. All compared methods, similarly to ENMIQA, do not require training. However, MEON and DEEPIQ represent recently introduced deep learning approaches and are already trained by their authors. The ENMIQA run with S=30 in experiments and other measures used their default parameters. In cases in which a method was designed to process color images, three identical channels were used as an input. The performance of the methods and their approaches to image quality modeling and prediction are shown in Table 2.

As reported, the measure introduced in this paper, ENMIQA, outperforms related techniques by a large margin in terms of all four performance indices. Depending on the considered index, it is followed by SISBLIM (PLCC and RMSE) and DEEPIQ (SRCC and KRCC). To show the performance of the measures for images of body parts largely represented in the database, the PLCC calculated for their subsets is reported in Figure 4. Here, ENMIQA obtains greater PLCC than it can be seen for the remaining methods for images of the lumbar and cervical spine, knee, shoulder, and wrist. It is slightly worse than BPRI for brain images. Interestingly, it seems that the recently introduced BPRI is suitable for such images, despite being the second worse technique regarding the entire database and the fourth-best technique in ranking based on the individual body parts. The worse results of methods designed for the assessment of natural images, as well as by complex deep learning approaches, can be justified by the specifics of MR images in which a large portion of the area is covered by organs or tissue while the background is usually dark and may contain noise. In natural images, such empty or nearly empty spaces are seldom found. Furthermore, popular BIQA methods are often trained to recognize typical distortion types (e.g., BPRI, ILNIQE, MEON, SSEQ, or DEEPIQ). Interestingly, methods trained on images contaminated with Gaussian noise can, to some extent, correctly predict the quality of MR images since Gaussian noise manifests itself in magnitude images as a Rician distribution of pixel intensities [38]. This is confirmed by weaker performance of the SNRTOI, which, being an SNR derivative, is often used by radiologists as supporting information on the captured images. The reported results for other methods seem to justify the need for the development of measures designed for the IQA of MR images.

To evaluate the statistical significance of the obtained errors in the prediction of IQA methods, hypothesis tests based on the prediction residuals of each IQA measure after non-linear mapping were conducted using F-statistic [28]. The F-test is based on an assumption of the Gaussianity of residuals and determines whether the two compared sample sets come from the same distribution, based on the ratio of their variances. The test is often used for the comparison of IQA measures [28]. Therefore, at first, the Jarque–Bera (JB) statistic to determine whether residuals come from a normal distribution was used [39]. In the JB test, the null hypothesis is that the vector of residuals of NR measure follows a normal distribution while the alternative hypothesis is that it does not follow it. Since for all compared measures the null hypothesis was not rejected at the 5% significance level, the F-statistic could be reliably employed. In the F-test, the null hypothesis is that the vectors of residuals of two IQA measures come from the same distribution with the same variance and are statistically indistinguishable (95% confidence). The alternative hypothesis is that the vectors are statistically distinguishable and have different variances. Before the calculation of the F-statistic, a vector of residuals of a measure was used to fit a normal distribution and 1000 samples were drawn from it. The tests revealed that the residual variance of ENMIQA is statistically smaller than those of all compared IQA methods with confidence greater than 95%. This is also indicated by the ratio in all cases. The obtained JB statistics for measures and ratios of the residual variances of algorithms to the ENMIQA are presented in Table 3.

### 3.4. Computational Complexity

The computational complexity of ENMIQA depends on the size of processed image (N×M), the length of the sequence of thresholds *S*, and the size of the neighborhood used for the NMS (k=3×3). Therefore, its computational complexity is of the order of O($NMSk^{2}$).

The introduced dataset was used to analyze the computational complexity of methods in terms of the average time taken to assess an image. The methods were run on a 2.2 GHz Intel Core CPU with 8 GB RAM using Matlab 2019b environment. Table 4 reports obtained timings. As shown, ENMIQA is slower than MEON, SINDEX, and SNRTOI, but it is faster than the remaining seven measures. The fastest methods (i.e., SINDEX and SNRTOI) are characterized by inferior IQA performance, and taking into account the results for more promising techniques, the introduced ENMIQA is relatively fast and provides the superior quality prediction of MR images.

### 3.5. Influence of Parameters

The ENMIQA is governed by the sequence of thresholds T=[1,2,⋯,S], $S\in Z^{+}$ used by the non-maximum suppression. Therefore, it is worth to determine how stable is its performance for various *S*. The *S* is the greatest threshold in the sequence and indicates its length. The PLCC performance of the method on the entire database, ranging *S* from 5 to 100 with the step of 5 is shown in Figure 5a. The previously introduced evaluation methodology was applied on the entire dataset to allow a coherent comparison with already reported results of other IQA methods (see Section 3.3). Considering the value of the threshold *S*, it can be set in between 20 and 60 without a visible drop in the prediction performance. Since ENMIQA exhibits a stable performance across the values of *S*, S=30 used in experiments is justified.

The non-maximum suppression selects a pixel with the extreme value, taking into account its eight neighbors and the threshold *t*. Since a pixel has 8 neighbors, it is reasonable to use its full neighborhood (the size of 8). However, the suppression can be modified to accept a lesser number of neighboring pixels that are used to indicate the local extrema (see Equation (Equation 2)). Therefore, in Figure 5b, the impact of the number of neighbors on the PLCC results of ENMIQA is shown. Here, if the number of used neighbors while determining the local extrema is lower than 8, the performance of the method visibly deteriorates. Hence, the entire pixel neighborhood should be considered by ENMIQA with the NMS. Interestingly, even with a smaller neighborhood the approach still offers a promising performance.

## 4. Conclusions

In this work, a new BIQA measure for the evaluation of MR images is proposed. The method uses the non-maximum suppression with a sequence of thresholds to detect local intensity extrema in MR images. A relationship between the number of extrema and entropy is investigated. Consequently, a new measure is introduced and experimentally validated against ten representative BIQA techniques on a database that contains MR images assessed by a large group of experienced medical professionals. The experimental comparison reveals that ENMIQA outperforms the-state-of-the-art measures by a large margin in terms of four performance criteria, confirming its suitability for the quality prediction of MR images.

To facilitate the replicability of the reported findings, as well as the applicability of the measure, the Matlab code of ENMIQA and the dataset are available at http://marosz.kia.prz.edu.pl/ENMIQA.html.

## Figures and Tables

**Figure 1 entropy-22-00220-f001:**
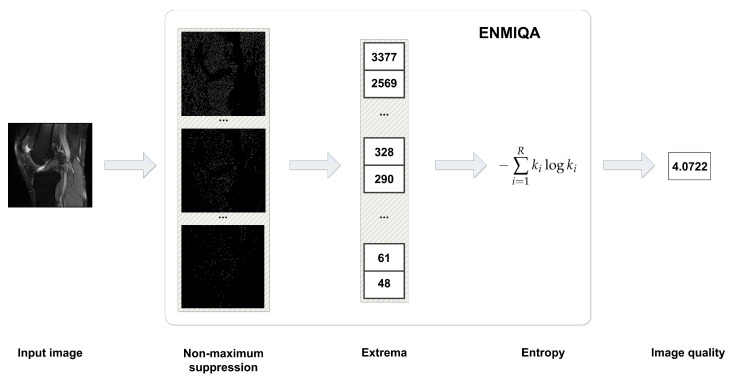
Image processing steps towards the calculation of image quality in ENtropy-based Magnetic resonance Image Quality Assessment measure (ENMIQA).

**Figure 2 entropy-22-00220-f002:**
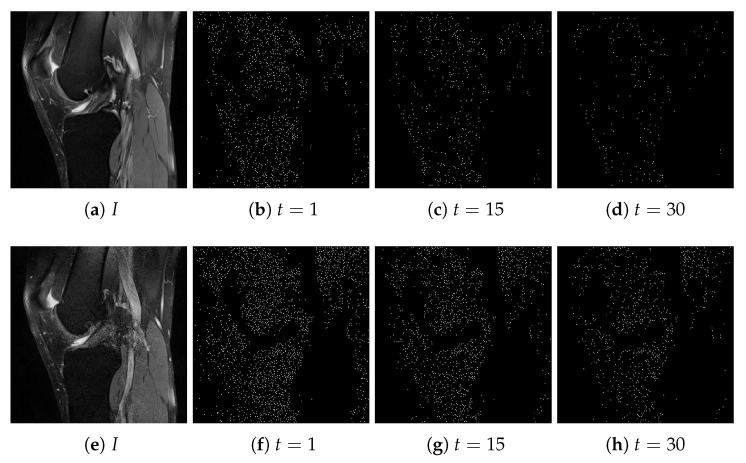
Two magnetic resonance (MR) images of different quality and the determined local extrema for t=1,15,30.

**Figure 3 entropy-22-00220-f003:**
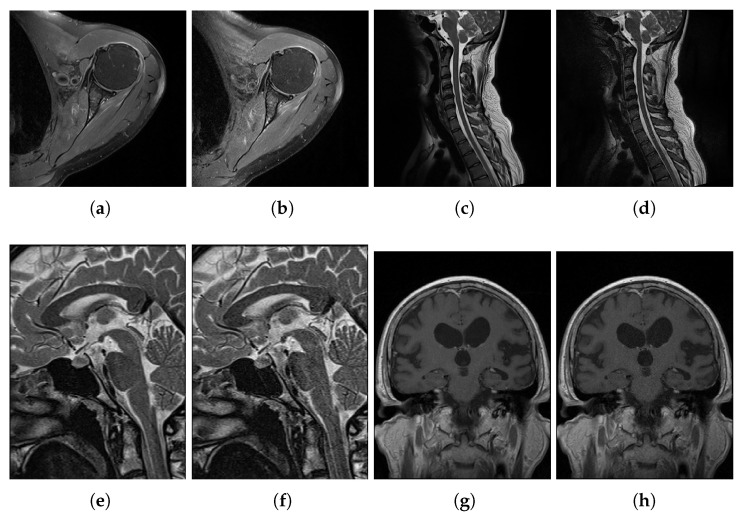
Exemplary MR images used in experiments.

**Figure 4 entropy-22-00220-f004:**
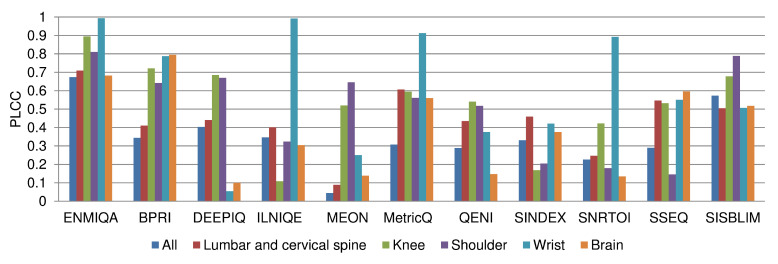
Pearson correlation coefficient (PLCC) performance of the BIQA methods for subsets of images of common body parts.

**Figure 5 entropy-22-00220-f005:**
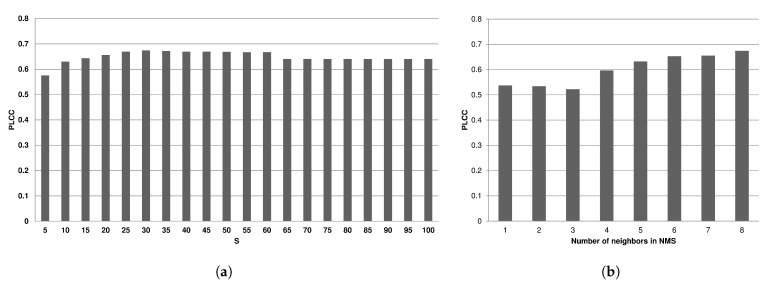
Influence of the threshold *S* (**a**) and the number of neighboring pixels in the non-maximum suppression (NMS) (**b**) on the PLCC performance of ENMIQA.

**Table 1 entropy-22-00220-t001:** Summary of images used in experiments.

Body Part	No. of Image Pairs	Axial Plane	Sagittal Plane	Coronal Plane
Lumbar and cervical spine	7	2	5	0
Knee	7	2	4	1
Shoulder	8	2	2	4
Wrist	3	0	0	3
Hip	2	1	1	0
Pelvis	2	0	0	2
Elbow	1	1	0	0
Ankle	1	0	1	0
Brain	4	1	2	1
Total pairs	35	9	15	11

**Table 2 entropy-22-00220-t002:** Evaluation and characteristics of compared blind image quality assessment (BIQA) measures. The best value for each performance criterion is written in bold.

Method	PLCC	SRCC	KRCC	RMSE	Approach to Image Quality Modeling and Prediction
ENMIQA	**0.6741**	**0.3540**	**0.2428**	**0.5375**	Thresholded NMS and entropy
BPRI	0.3440	0.1515	0.1120	0.6832	Distortion-specific metrics and pseudo-reference image
DEEPIQ	0.4039	0.3030	0.2037	0.6657	RankNet trained on quality-discriminable image pairs
ILNIQE	0.3465	0.1796	0.1162	0.6826	Multivariate Gaussian model of pristine images
MEON	0.0439	0.1247	0.0771	0.7272	End-to-end deep neural network with subtasks
MetricQ	0.3075	0.2300	0.1520	0.6924	Singular value decomposition of local image gradient matrix
QENI	0.2886	0.2385	0.1587	0.6967	Self-similarity of local features and saliency models
SINDEX	0.3307	0.2802	0.1962	0.6869	Global and local phase information
SNRTOI	0.2262	0.1828	0.1245	0.7088	Signal-to-nose ratio
SSEQ	0.2903	0.0855	0.0487	0.6963	Distortion classification using local entropy
SISBLIM	0.5733	0.2885	0.1820	0.5962	Free energy theory based fusion of distortion-specific metrics

**Table 3 entropy-22-00220-t003:** Ratios of residual variances of methods to ENMIQA and the Jarque–Bera (JB) statistics. Smaller values of JB statistics denote smaller deviations from the Gaussianity. All measures follow a normal distribution.

Method	Ratio	JB Statistic
ENMIQA	1.0000	0.8523
BPRI	0.6189	2.8999
DEEPIQ	0.6510	1.3870
ILNIQE	0.6201	3.9911
MEON	0.5462	3.8930
MetricQ	0.6032	2.8356
QENI	0.5952	2.7040
SINDEX	0.6124	3.2580
SNRTOI	0.5751	1.7389
SSEQ	0.5958	3.5343
SISBLIM	0.8128	0.1254

**Table 4 entropy-22-00220-t004:** Time–cost comparison of BIQA measures (in seconds).

Method	ENMIQA	BPRI	DEEPIQ	ILNIQE	MEON	MetricQ	QENI	SINDEX	SNRTOI	SSEQ	SISBLIM
**Runtime**	0.2151	0.2524	2.439	9.299	0.1853	0.4813	1.212	0.0479	0.0069	0.9140	1.629

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
