# Peer review of "Magnetic Resonance Image Quality Assessment by Using Non-Maximum Suppression and Entropy Analysis"

_entropy, 2020, doi:10.3390/e22020220_

Round 1

Reviewer 1 Report

Summary

This manuscript introduces a non-reference (blind)image quality assessment (BIQA) method for the automatic evaluation of magnetic resonance (MR) images. The analysis pipeline is based detecting local intensity extrema in MR  by utilizes the non-maximum suppression (NMS) with a sequence of thresholds. The framework has been tested on a cohort of 51 patients  that contains T2-w MR images and subjective scores collected in tests with human subjects. The presented pipeline demonstrated the capability for the quality prediction of MR images based on using Pearson correlation coefficient (PLCC), Spearman Rank order Correlation Coefficient (SRCC), Kendall Rank order Correlation Coefficient (KRCC), and Root Mean Square Error (RMSE).. Additionally, the experimental results documented the high outperformance of the presented approach over other state-of-the-art techniques.

In general, the paper investigates an important MR application that fits the journal readers. Also, the reference list is composed of up-to-date literature work. However, it lacks theoretical contribution and the paper language is somehow hard to follow.

Contribution /Novelty

First, the algorithms used within the pipeline are already published work, however this has been compensated for by the adaptation of those approaches and integrate them to serve for the study.

Major Strengths

Novel idea, Important application, and well-defined problem Comparison with stat-of-the-art approaches

Major Criticisms

Marginal theoretical contribution Statistical analysis is missing

Detailed Comments

Abstract section needs to be re-written. Instead of only providing a synopsis of the paper, it would be good if the authors provide a brief view on the need/importance for the research work. Also, it lacks any quantitative results. The authors statements about FR IQA, on page 2 lines 23-29 are not supported by references. The language of the text is hard to follow. The authors use very long sentences (e.g., lines 33 till lines37 is a single sentence) which confuses the reader. Also, the abbreviations style is not consistent. Please compare, for example, NQM (Page 2 line 34) with MSE (page 2 line 31). Thus, the manuscript is suggested to be checked by a native speaker of the English language or by anyone authorized to do the same. Most of the employed methods are already published work and the authors’ theoretical contribution, in my opinion, is marginal. The idea of using Entropy is however interesting and the experimental comparison with other techniques is promising based on the reported results. Although the comparison with other approaches are employed and documents the superiority of the paper, the statistical analysis is missing. Thus, it is very hard to draw a superiority conclusion. For the compared methods, did the authors used available software or they reimplemented the methods? If it is the later, how about parameter tuning? Please discuss According to the data presented in Fig. 5, the best neighborhood size is at 8. When comparing Fig 5(a) with Fig 5 (b), there is trend of the figure 5 (b), i.e., increasing the neighborhood size the performance also increase. A discussion is need here. Was the influence of the parameters on the system performance was conducted on specific type of images? Or all images? Please add details. I am not sure how the authors calculated the score of the processed images and compare it with the MOS

Reviewer 2 Report

Objective and automated estimation of medical image quality is full of practicality both in clinic and industry fields. In this paper, the authors propose a no-reference image quality assessment method and evaluated the method on T2 weighted magnetic resonance images. This work is interesting.

For further improve the manuscript quality, several issues should be well tackled.

1. Experimental data requires more details. It also needs some tables or figures to demonstrate the data processing. Therefore, please follow some clinical papers to describe these details.

(a) Indeed, how many MR images are collected in the database? There are several numbers confusing, “a group of 51 patients”, “since a database for IQA should contain images of a different quality”, “there are 30, 18, and 22 images captured in sagittal, axial, and coronal planes”. Please check these numbers and summarize in a table.

(b) How to score these MR images ? Is each image scored by all 31 radiologists ? Please clarify it.

(c) How to obtain the MOS ? Please follow the reference [27] and enrich the flowchart. Please keep in mind that your paper is an independent paper. Readers should know these technical details and if possible, they prefer not to read other papers.

(d) Privacy issue of clinical data. Were the patients formally informed the use of these images? Was written or verbal consent obtained ? Please specify it.

(e) According to Figure 4, please specify how many images with regard to different organ? A table is suggested to show these information.

2. The proposed method is incomplete. Both the operation of non-maximum suppression and the calculation of entropy are introduced. However, how to build the relationship between the entropy and the image quality score? Please specify this key point.

3. Comparative evaluation.
(a) It is better to describe these metrics briefly and show the principles to readers. Then, the readers could understand and decide whether these methods are suitable to tackle this problem of blind medical image quality assessment.

(b) If possible, figure out whether some of these methods require learning on medical images.

(c) If possible, please show some underlying reasons why “the recently introduced BPRI is suitable for such images, despite being the second worse technique regarding the entire database and the fourth-best technique considering ranking based on the individual body parts.”, and why other methods fail on this data set?

4. Please highlight the importance of medical image quality assessment (Introduction part). There are 31 radiologists and whether they could give some reasons why medical image quality assessment is important, what is the difference between medical and natural image quality assessment ? any potential applications of medical image quality assessment ? Please ask widely and think deeply and then summarize these information. These messages are important to the readers when the field of medical image quality assessment is at its beginning.

5. The manuscript should be proof-read by native speaker due to unclear expression.

For instance,
(a) Title: How about “Magnetic resonance image quality assessment by using non-maximum suppression and entropy analysis” or “Image quality assessment of T2-weighted magnetic resonance imaging via non-maximum suppression and entropy analysis” ?

(b) Line 002-003: “In the method, pixels of the assessed image are subjected to the non-maximum suppression (NMS) operation varying the level of acceptable local intensity differences.” How about “In the method, an image is first processed by non-maximum suppression (NMS) with various levels of acceptable local intensity difference.”

Round 2

Reviewer 1 Report

The authors have accounted for all the reviewers comments. No more revisions are needed

Reviewer 2 Report

The comments are well addressed. Due to the low performance on the dataset (PLCC<0.6741 and SRCC <0.3540), I think the authors should give some insights or clues on the further improvement of their work.

Minor comments

How to further improve your work on the dataset for blind medical image quality assessment? Please discuss this issue in the Discussion part. To Figure 5 (b), it is common to know one pixel has 4 or 8 connected neighbors. I am wondering how to define its 2 or 5 connected neighbors? Please specify. The manuscript should be further proof-read due to unclear expression.

For instance,

Line 207-208, “Here, if the number of used neighbors while determining the local extrema is lower than 8, the performance of the method visibly deteriorates”?